# A Carrier Female Manifesting an Unusual X-Linked Retinoschisis Phenotype Associated with the Pathogenic Variant c.266delA, p.(Tyr89LeufsTer37) in *RS1*, and Skewed X-Inactivation

**DOI:** 10.3390/genes14061193

**Published:** 2023-05-29

**Authors:** Jennifer Kirkby, Stephanie Halford, Morag Shanks, Anthony Moore, Anthony Gait, Lucy Jenkins, Penny Clouston, Chetan K. Patel, Susan M. Downes

**Affiliations:** 1Oxford Eye Hospital, John Radcliffe Hospital, Oxford University Hospitals NHS Foundation Trust, Oxford OX3 9DU, UK; jennifer.kirkby@nhs.net; 2Nuffield Laboratory of Ophthalmology, Nuffield Department of Clinical Neuroscience, University of Oxford, Level 6 John Radcliffe Hospital, Headley Way, Oxford OX3 9DU, UK; stephanie.halford@eye.ox.ac.uk; 3Oxford Medical Genetics Laboratory, Oxford University Hospitals NHS Foundation Trust, Oxford OX3 9DU, UK; 4UCL Institute of Ophthalmology, University College London, London EC1V 9EL, UK; 5Moorfields Eye Hospital NHS Foundation Trust, London EC1V 2PD, UK; 6Department of Ophthalmology, UCSF School of Medicine, San Francisco, CA 94158, USA; 7Rare & Inherited Disease Genomic Laboratory, Great Ormond Street for Children NHS Foundation Trust, London WC1N 3JN, UK

**Keywords:** X-linked retinoschisis, heterozygous, female, carrier, X-inactivation

## Abstract

X-linked retinoschisis (XLRS) is the most common juvenile macular degeneration in males. Unlike most other X-linked retinal dystrophies, carrier heterozygous females are very rarely reported to show clinical features of the disease. Herein, we describe unusual retinal features in a 2-year-old female infant with family history and genetic testing consistent with XLRS.

## 1. Introduction

X-linked retinoschisis (XLRS) is a retinal dystrophy characterised by schisis, or splitting, of the neural retina. The causative gene, *RS1*, located at Xp22.1, was identified in 1997 [1] and encodes the protein RS1. RS1 is important in maintaining the structural integrity of the retina and numerous loss of function variants have since been described [2]. XLRS is the leading cause of juvenile macular degeneration in males [3], with a reported incidence of 1 in 15,000 to 1 in 30,000 [4].

There is a wide range of phenotypic variability both inter- and intrafamilially even with the same causative *RS1* variant [4]. Best corrected visual acuities may vary from 20/20 to 20/600 [3]. Clinical diagnosis of XLRS can be challenging, with reports of an average delay of 8 years from symptom onset [5]. The hallmark foveal schisis has been described as the most common abnormality in those under 40 years old, though it may become less distinct with time and may not be present at all [6,7]. More than half of patients have peripheral retinoschisis [5], particularly in the inferotemporal region; in infancy, these may take the form of bullous retinoschisis, which generally regresses, leaving pigment lines [8]. Retinal detachment may complicate cases seen in peripheral retinal breaks, bilateral macular detachments, abnormal vitreomacular traction and macular holes [9]. Unsupported bridging vessels between layers of schisis may be subject to shearing force leading to vitreous haemorrhages.

There are rare reports of female XLRS cases [7,10,11,12,13,14,15,16,17,18,19,20,21,22,23] and a review of these is found in our discussion. Whilst classically, XLRS heterozygous carrier females are considered asymptomatic with no clinical features [24], there are reports to the contrary in the literature [7,10,11,12,13,14]. Furthermore, we describe a young female with retinal findings in whom genetic analysis has provided a molecular diagnosis, indicating a rare heterozygous state and X-inactivation.

## 2. Case Report

A 2-year-old female was referred to a tertiary eye hospital; her parents had noted that she was bumping into objects on her right-hand side; no nyctalopia was observed. A right esotropia was seen by her parents at 2 weeks of age. She was born at 39 weeks’ gestation by spontaneous vaginal delivery. In the neonatal period she was treated for presumed sepsis but was otherwise fit and well. Her cycloplegic refraction was right +2.50/−0.75 × 180 and left +1.00/−1.00 × 180.

Her 30-year-old father had poor vision since childhood with a history of nystagmus and early onset esotropia and was registered as severely sight impaired. He did not have any formal clinical diagnosis and had never undergone any previous genetic analysis; however, the referring hospital had noted bilateral foveal schisis. There was no other family history of any ocular conditions nor reduced vision, including the proband’s mother and grandparents. There was no family history of consanguinity.

Examination under anaesthetic (EUA) of the proband was performed at 2 years old and repeated at the age of 3 years, 7 months old. Fundus imaging was performed using Retcam; fundus autofluorescence, fundus fluorescein angiography and OCT imaging were captured with Heidelberg HRA+OCT Spectralis.

On examination, there was no iris hypoplasia. Fundoscopy revealed (Figure 1A,B) an area of thinned elevated retina in the left inferior periphery; within this area, a circular outer leaf break was evident on fundus fluorescein angiography (FFA) (Figure 1C) and OCT imaging of this area confirmed it as inferior schisis (Figure 1E,F).

A well circumscribed area of abnormal retina was observed in the right inferotemporal periphery (Figure 2); this comprised RPE atrophy peripherally, within which two concentric rings of pigment enclosed a schitic appearing area with associated yellow material. OCT imaging through this lesion did not demonstrate any schisis (Figure 2D,E).

Both maculae appeared normal on examination, but macular OCT imaging revealed thickened retinae throughout all layers (Figure 1D). In addition, bilateral mild foveal hypoplasia was observed with broadened foveal dip. There was an absence of laminal structure within the thickened retinae and minute intraretinal hyporeflective cystic lesions (Figure 1D, arrows) were observed.

In the interval between the two EUAs, there were no new changes and no evidence of progression.

Electrodiagnostic testing under general anaesthetic showed clearly defined electroretinograms in both scotopic (Figure 3A) and photopic (Figure 3B) ambient conditions with reproducible implicit times. Waveforms produced from both eyes were not electronegative, though the b wave amplitude was reduced in the right eye relative to her left (photopic b/a ratios 1.8 right and 2.8 left eye). Flash VEPs were not discernible, presumably due to the effect of Sevoflurane (an inhalational anaesthetic).

The opportunity was taken to examine the proband’s father. Optos wide field retinal imaging (Figure 4A,B) showed bilateral macular retinal pigment epithelial changes, and vitreous veil at the midperiphery. Macular OCT imaging demonstrated a central hyporeflective intraretinal cavity in the right eye (Figure 4C) with adjacent smaller cystic-like intraretinal lesions peripherally; in the left eye, there were similar large cystic-like hyporeflective intraretinal lesions throughout, maximal in size and coalescing centrally (Figure 4D).

DNA obtained from a venous blood sample from the proband was sent for targeted sequencing by the Oxford Medical Genetics Laboratory using their next generation sequencing (NGS) phenotype panels. The coding exons and at least 10 bp of the flanking introns of 111 retinal genes were captured as previously described [25]; the target genes and NGS coverage data are provided in the Appendix A (Appendix A). This identified a heterozygous variant in *RS1* (c.266delA, p.(Tyr89LeufsTer37)) at ChrX g.18,655,371 based on build GRCh37/hg19 and nucleotide and protein numbering according to *RS1* transcript NM_000330.3. To the best of our knowledge, this is a novel variant. The single nucleotide deletion causes a frameshift that results in premature termination of the protein. This confirmed a molecular diagnosis of X-linked retinoschisis in the proband.

The proband’s father underwent molecular genetic analysis, as described above, and the same variant was identified. This not only provided him with a molecular diagnosis of X-linked retinoschisis, but also indicated paternal inheritance of the variant in the proband.

X-inactivation testing of the proband was performed in an accredited National Health Service (NHS) laboratory using DNA extracted from a venous blood sample (standard DNA extraction using Promega, Tecan Freedom EVO^®^ HSM Workstation), and methylation analysis of the androgen receptor locus. This method utilises polymorphic markers within the locus which differentially methylate on the inactive and active X chromosomes, which are then analysed by fluorescent PCR in the presence and absence of the methylation sensitive restriction enzyme *Hpa*II. X-inactivation testing revealed a skewed pattern of X-inactivation with an approximate ratio of 86:14.

## 3. Discussion

We report a case of XLRS in a female child with retinal findings in keeping with a severe phenotype, in whom genetic analysis has provided a molecular diagnosis with a heterozygous variant in *RS1* (c.266delA, p.(Tyr89LeufsTer37)) and skewed X-inactivation.

To date, there are fifteen reports describing 30 cases of XLRS in females in the literature [7,10,11,12,13,14,15,16,17,18,19,20,21,22,23] (see Table 1 for summary). The majority of cases (19 of 30) have been reported in families involving consanguineous unions [15,16,17,18,19,20,21,22]. They are either presumed, or demonstrated on molecular genetic analysis, to be homozygous or compound heterozygous for *RS1* variants. Another reported case is of an affected female with Turner’s syndrome and no family history of consanguinity [23].

Ten cases of females with XLRS have been reported in the heterozygous (carrier) state [7,10,11,12,13,14]. In the majority of these reports, carrier status was inferred, without genetic analysis. Two early reports described the appearances of obligate carrier females in XLRS pedigrees: a 41-year-old female with bilateral inferotemporal retinoschisis, and an asymptomatic 13-year-old with 20/25 vision, bilateral small central scotomas and left macular “pigment mottling” changes [10,11]. Wu et al. described a 56-year-old obligate carrier with unilateral “radial wrinkling of the internal limiting membrane around the fovea”, though no evidence of foveal schisis in either eye [12]. Kaplan et al. suggested features of XLRS may occur more frequently in female carriers than previously accepted [13]. Their report examined five obligate carriers and, through genetic studies with polymorphic probes on the distal short arm of the X chromosome, were able to confirm carrier status. Although none showed macular alterations, four had peripheral retinal changes, which ranged from “mild greyish-white spots or dendrite-like areas”, “aberrant zones of underdeveloped capillaries”, to inferotemporal peripheral retinoschisis; two had a history of childhood cryocoagulation for undescribed peripheral lesions. Ali et al. described a 54-year-old obligate carrier with inferior retinoschisis, and an outer retinal hole with lattice degeneration in one eye [14].

The case we report here presented in early childhood, and in contrast to the above heterozygous cases, demonstrates a more severe phenotype. Whilst the typical foveal schisis of XLRS was not present, our case showed a thickened retina and minute hyporeflective cysts on OCT imaging which may represent subtle schitic changes with the potential to progress to a more typical picture in time. The lack of macular schisis has been reported before [6] and in a female [7]. The inferior schisis located in the left eye is in keeping with typical XLRS. The right inferotemporal island of abnormality is somewhat unusual because it is unilateral, there is no clear schisis on OCT imaging, and it is surrounded by normal retina. Usually, in XLRS, a peripheral schisis would be seen to extend to the ora serrata. However, XLRS is notoriously variable in phenotype and bullous detachments in infants have been reported and shown to regress leaving pigment lines [8] as described surrounding the island of abnormality in the presented case. The associated yellow material is presumed to be lipid exudation which can be found in XLRS; there was no evidence of any associated telangiectasis. The macular OCT appearances might have suggested the possibility of enhanced S-cone syndrome but the history and overall features were not typical; there was no band of pigmentary change and the electrophysiology was not in keeping with *NR2E3* retinopathy [26].

The molecular genetic analysis was performed in one of three accredited genomic centres in the UK National Health Service (NHS) specialising in ocular disease. The analysis performed on the proband identified a single pathogenic variant in *RS1* (c.266delA, p.(Tyr89LeufsTer37)) in an heterozygous state. This single nucleotide deletion causes a frameshift leading to premature termination of the protein. No other potentially disease-causing variants were detected. No variants of uncertain clinical significance (VUS) were reported. No polymorphisms were identified in *RS1*; polymorphisms were found in other genes on the 111 panel, including in *NR2E3*, but according to the Association of Clinical Genetic Science guidelines, they were not considered to be pathogenic [27]. Analysis for deep intronic variants (i.e., >100 bp away from the intron/exon boundary) was not undertaken given the healthcare setting where this is not routinely available. To date, there is no evidence in the literature for the existence of deep intronic variants in XLRS. The *RS1* variant, also present in the father, was reported to be pathogenic by the NHS laboratory. The proband’s mother, who was asymptomatic, did not undergo testing as she did not consent at the initial meeting. X-inactivation testing revealed a skewed pattern of X-inactivation with an approximate ratio of 86:14.

Interestingly, an extensive literature search uncovered the report of a similar case [7]. Saldana et al. describe a female with clinical signs of XLRS who was heterozygous for her affected father’s *RS1* variant (c.305G > A; p.(Arg102Gln)). She presented at 5 years old with poor vision (20/30 and 20/40 best corrected visual acuity). Analogous to the presented case, there are mild macular signs with no foveal schisis; retinal pigment epithelial changes are described, though OCT imaging is not included. Examination of the maculae in the presented case was normal, with abnormalities only evident on OCT imaging, including minute intraretinal hyporeflective cystic lesions. Saldana et al. [7] additionally report bilateral inferotemporal peripheral retinoschisis and right inferonasal peripheral retinoschisis. The proband in our presented case similarly had peripheral retinischisis inferiorly in the left, though a relatively unusual right inferotemporal lesion as described.

The remaining nineteen cases of female XLRS are discussed for completeness, especially given the paucity of reports, and are all members of consanguineous family pedigrees. They are therefore either presumed, or demonstrated on molecular genetic analysis, to be homozygous or compound heterozygous for *RS1* variants. Ali et al. reported a family with four affected sisters, an affected father, and clinically normal brother and carrier mother [15]; three of the sisters had bilateral macular schisis whilst the third had bilateral retinal detachments, visual acuity ranged from 6/36 to LP. Rodriguez et al. described three affected females from a large Colombian family, homozygous for the *RS1* c.639delG variant [16]; all six eyes displayed macular schisis of the macula, whilst peripheral inferotemporal retinoschisis was found in five. Saleheen et al. [17] reported severe XLRS phenotypes in four sisters who were homozygous for a novel c.579dupC variant in exon 6 of the *RS1* gene; all presented early in life, with one presenting at 1 year old with bilateral retinal detachment and retinoschisis with macular involvement; a 3-year-old had bilateral complete retinal detachments. The two other sisters had typical macular schisis in a stellate pattern, with one additionally having bilateral inferior scars with pigmentary changes accompanying peripheral retinoschisis. Gliem et al. [18] described a family all with the same missense variant, c.293C > A, p.(Ala98Glu), in exon 4 of the *RS1* gene. The 59-year-old homozygous mother had reported reduced visual acuity since childhood, and seven years prior to diagnosis, had underwent cryotherapy for suspected bilateral Coats disease; she had an atrophic macula (with mottled reduced autofluorescence signal, surrounded by a ring of increased autofluoresence) and hyperpigmentary changes and chorioretinal scarring towards the periphery. Saffieri et al. [19] identified two affected sisters (an 11-year-old and 9-year-old) as homozygous for a known disease-causing variant c.304C > T (rs61752067). Both presented in infancy with sensory nystagmus, high hypermetropia, and nyctalopia, though only the elder sister had typical XLRS features of schisis. Khan and El-Ghrably [20] describe a female homozygous for the known pathogenic *RS1* variant c.304C > T; p.(Arg102Trp), with poor vision since childhood and previous “laser” treatment to her left eye. She had characteristic features of bilateral central and peripheral neurosensory retinal splitting in a classic pattern. Önen et al. [21] investigated a pedigree with three affected sisters, aged from 4 to 17 years old, with visual acuity range from 20/40 to 20/100. The two younger sisters had better vision and foveomacular retinoschisis, whilst the eldest had an atrophic macula with pigment epithelial changes (appearances similar to their father who reported poor vision since childhood); their presumed carrier mother and sister were unaffected. Another study reported an 18-year-old girl with counting fingers vision, poor vision since childhood and bilateral macular holes, who was found to have an unspecified *RS1* variant [22].

In these cases, where both alleles are affected, the retinal features are more consistent with typical XLRS, though wide variation in phenotype is evident, including within those affected by the same causative variant. Of interest, relating to the island of abnormal retina (Figure 2) in our reported case, Saleheen et al. [17] described a 5-year-old with bilateral inferior scars with pigmentary changes located in a similar midperipheral position, and Gliem et al. [18] presented a 59-year-old showing, amongst other signs, hyperpigmentary changes and chorioretinal scarring towards the periphery.

For those in whom only one *RS1* variant is present, as seen in our case, and those described by Saldana et al. [7] and others [10,11,12,13,14], one explanation for the variability in phenotype could be skewed X-inactivation. Saldana explored this in their case but studies on X-inactivation using triplet repeat markers were uninformative.

X-inactivation occurs early in embryonic development and is the mechanism by which a random X-chromosome undergoes random epigenetic silencing. The resultant X-chromosome inactivation (XCI) ratio in newborn females follows a normal distribution with a peak at 50:50, reflecting the randomness of choice. If a bias towards a particular X-chromosome occurs, this XCI ratio will be skewed. Examples of biases include influences on the X-inactivation process (such as X-linked variants in the mediator *Xist* [28]), and secondary skewing, whereby a disease-causing X-linked variant may affect cell survival and replication during development, relative to wildtype. At the extremes of distribution where there is significant skewing, female carriers of X-linked recessive variants may manifest disease through biased XCI in favour of the disease-causing variant. It has been postulated that such skewing may explain the variable manifestations of X-linked retinitis pigmentosa in female carriers [29]. Additionally, there exists a subgroup of X-linked genes with the ability to escape XCI and continue to be expressed from both chromosomes, such as *RP23* [30]; *RS1* has been found to have “variable escape” between individuals [29].

Skewing additionally occurs at later stages of development and in life; XCI ratios skew with age and in some tissue more than others [29]. Blood is particularly susceptible to increased skewing over time and may not accurately reflect XCI ratios in the retina [31]. However, the relative ease of accessibility means blood remains the most frequently sampled tissue in the literature for determining XCI ratio.

X-inactivation testing analysis in the case we report here did show a skewed pattern of X-inactivation, which could be the explanation for the manifesting phenotype in this individual.

## 4. Conclusions

In conclusion, we present a young female with retinal findings in keeping with a severe XLRS phenotype, as well as some unusual features, in whom genetic analysis has provided a molecular diagnosis, indicating a rare heterozygous state and X-inactivation. It is important to consider XLRS in females where the features may not be typical and consider X-inactivation studies in these rare cases, particularly as there may be the potential for treatment in the future.

## Figures and Tables

**Figure 1 genes-14-01193-f001:**
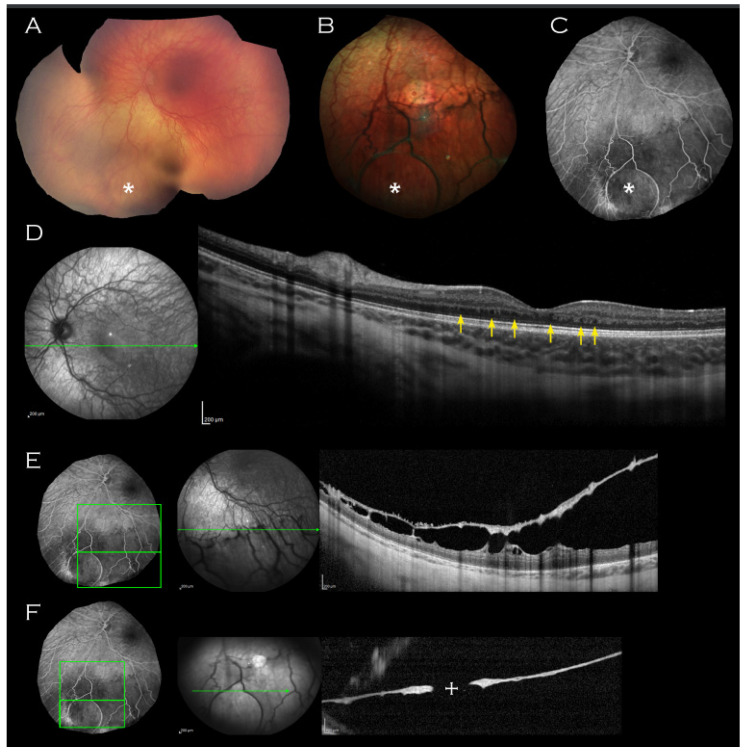
Left eye imaging of proband. (**A**) Retcam montage, (**B**) colour fundus photograph and (**C**) fluorescein angiogram shows circular area of thinned elevated retina (indicated by asterisk) seen consistently on all imaging modalities. (**D**) Macular OCT demonstrating unusual thickening of the retina with absence of laminal structure and minute intraretinal hyporeflective cystic lesions (indicated by arrows) and (**E**,**F**) OCT through the lesion with approximate mapping of location on fluorescein angiogram image. An associated outer leaf break is demonstrated ((**F**), cross).

**Figure 2 genes-14-01193-f002:**
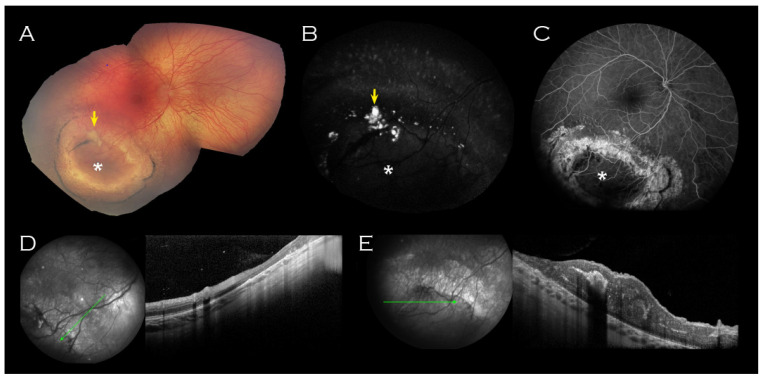
Right eye imaging of proband. Inferotemporal peripheral island of abnormal retina (indicated by asterisk) on (**A**) Retcam montage, (**B**) fundus autofluorescence and (**C**) fluorescein angiogram. Associated yellow material (arrow (**A**,**B**)) was hyperfluorescent on fundus autofluorescence. (**D**) OCT of the surrounded atrophic retina corresponded with thinned retina and (**E**) OCT through the centre of the lesion showed thickened retina corresponding to the relatively hypofluorescent island seen on FFA. No schisis was demonstrated on OCT through the lesion.

**Figure 3 genes-14-01193-f003:**
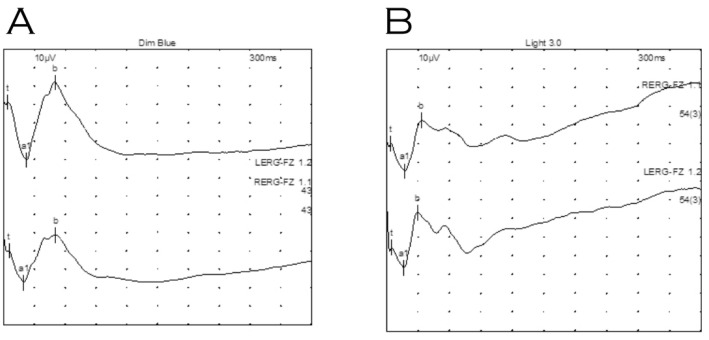
Electroretinography of proband under general anaesthetic. (**A**) Dark adapted dim blue scotopic response. (**B**) Light adapted standard flash (LA 3.0). Waveforms were not electronegative, however, waveforms for right eye (top) indicate relatively reduced b wave amplitudes compared to those for the left eye (bottom).

**Figure 4 genes-14-01193-f004:**
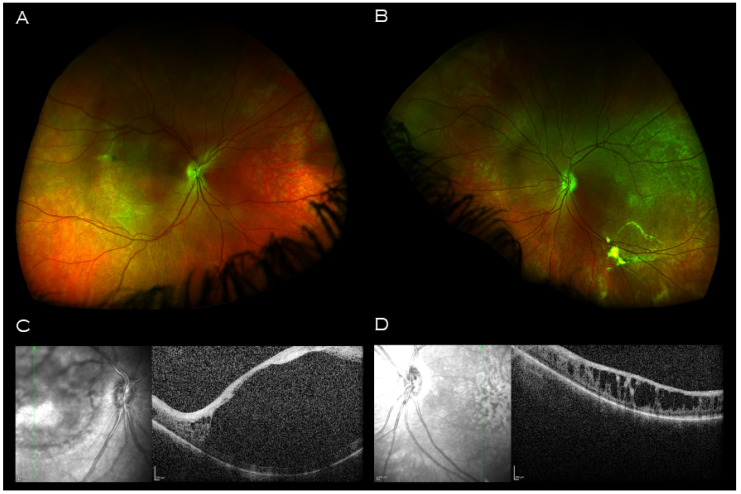
Imaging of proband’s father. Wide field retinal imaging (**A**) right eye (**B**) and left eye shows mild retinal pigment epithelium changes at the maculae and left vitreous veil at left inferior arcade. (**C**) Macular OCT right eye demonstrating a central hyporeflective intraretinal cavity with adjacent small cystic-like hyporeflective intraretinal lesions peripherally. (**D**) Macular OCT left eye demonstrating large cystic-like hyporeflective intraretinal lesions throughout, maximal in size and coalescing centrally.

**Table 1 genes-14-01193-t001:** Summary of reported female XLRS cases in the literature (BCVA, best correct visual acuity; CF, count fingers; LP, light perception).

Authors	Age	Zygosity	*RS1* Variant	BCVA	Macular Findings	Retinal Findings
Kirkby et al. 2023	2 years old	Heterozygous	c.266delA, p.(Tyr89LeufsTer37)	Unavailable	Thickened retinae with absence laminal structure. Mild foveal hypoplasia. Minute intraretinal hyporeflective cystic lesions; no foveal schisis.	Inferior peripheral retinoschisis (left). Inferotemporal RPE atrophy (right, with associated concentric rings of pigment enclosing a schitic-appearing area and yellow material).
Saldana et al. [7]	5 years old	Heterozygous	c.305G>A; p.(Arg102Gln)	20/30 and 20/40	RPE changes; no foveal schisis.	Bilateral inferotemporal peripheral retinoschisis.
Gieser et al. 1961 [10]	13 years old	Obligate carrier	Unavailable	20/25 (Central scotomas)	Foveal schisis with “pigment mottling and few verrucae in sharply punched-out macular area.”	
Sabates 1966 [11]	41 years old	Obligate carrier	Unavailable	Unavailable		Temporal (right) and inferotemporal (left) retinoschisis. Two areas midperipheral “pigment clumping.”
Wu et al. 1985 [12]	56 years old	Obligate carrier	Unavailable	Unavailable	Radial wrinkling around fovea, internal limiting membrane; no foveal schisis	
Kaplan et at 1991 [13]	Unavailable	5 x Obligate carriers	Unavailable	Unavailable	None had macular findings	Peripheral lesions (“mild greyish-white spots or dendrite-like areas,” “aberrant zones of underdeveloped capillaries,” inferotemporal retinoschisis). Two had previous childhood cryocoagulation.
Ali, S et al. 2013 [14]	54 years old	Obligate carrier	Unavailable	Unavailable		Inferior retinoschisis, outer retinal hole with lattice degeneration in one eye.
Ali, A et al. 2003 [15]	Proband (unavailable)	Presumed homozygous or compound heterozygous	Unavailable	LP (Bilateral nystagmus)		Bilateral total retinal detachment.
	10 years old	Presumed homozygous or compound heterozygous	Unavailable	6/60 and 3/60	Foveal schisis.	Bilateral longstanding inferior retinoschisis extending to inferior arcades.
	5 years old	Presumed homozygous or compound heterozygous	Unavailable	6/36 and 6/60	Foveal schisis with pigmentary changes.	Bilateral inferior scars and retinoschisis with pigmentary changes.
	1 year old	Presumed homozygous or compound heterozygous	Unavailable	Unavailable	“Macular involvement” of retinoschisis.	Bilateral retinal detachment, and retinoschisis. Secondary changes in areas of detachment, markedly atrophic retinae.
Rodriguez, F.J. et al., 2005 [16]	10 years old	Homozygous	c.639delG	20/20 and 20/30	Foveal schisis.	Mid-retinal cyst inferotemporal quadrant. Pale optic discs.
	37 years old	Homozygous	c.639delG	20/100 and 20/400	“Modified” foveal schisis	Previous cryotherapy inferotemporally. Pale optic discs.
	37 years old	Homozygous	c.639delG	20/60 and 20/400	“Modified” foveal schisis	Peripheral retinoschisis. Pale optic discs.
Saleheen, D. et al. 2008 [17]	1 year old	Homozygous	c.579dupC	Unavailable	“Macular involvement” of retinoschisis.	Bilateral retinal detachment.
	3 years old	Homozygous	c.579dupC	LP (Bilateral nystagmus)		Bilateral complete retinal detachment.
	10 years old	Homozygous	c.579dupC	6/60 and 3/60	Foveal schisis radiating to inferior arcades.	
	5 years old	Homozygous	c.579dupC	6/36 and 6/60	Foveal schisis.	Bilateral inferior scars with pigmentary changes accompanying peripheral retinoschisis.
Gliem, M. et al. 2014 [18]	59 years old	Homozygous	c.293C>A, p.(Ala98Glu)	Unavailable	Atrophic macula.	Hyperpigmentary changes and chorioretinal scarring towards periphery. Prior cryotherapy.
Staffieri, S.E. et al., 2015 [19]	9 years old	Homozygous	c.304C>T (rs61752067)	Sensory nystagmus		
	11 years old	Homozygous	c.304C>T (rs61752067)	Sensory nystagmus	Foveal schisis.	
Khan, A.O. et al., 2019 [20]	Unavailable	Homozygous	c.304C>T p.(Arg102Trp)	“Poor since childhood”	Foveal schisis	Peripheral retinoschisis.
Onen, M. et al. 2020 [21]	4 years old	Presumed homozygous or compound heterozygous	Not specified	20/50	Foveal schisis.	
	15 years old	Presumed homozygous or compound heterozygous	Not specified	20/40	Foveal schisis.	
	17 years old	Presumed homozygous or compound heterozygous	Not specified	20/100	Atrophic macula with RPE changes.	
Altun, A. et al. 2020 [22]	18 years old	Presumed homozygous or compound heterozygous	Not specified	CF	Bilateral macula holes.	
Sato, M. et al. 2003 [23]	29 years old	Turners syndrome	c.(522+1G>A)	60/200	Foveal schisis (diagnosed at 10 years old).	

## Data Availability

Not applicable.

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
