# Peer review of "A Carrier Female Manifesting an Unusual X-Linked Retinoschisis Phenotype Associated with the Pathogenic Variant c.266delA, p.(Tyr89LeufsTer37) in RS1, and Skewed X-Inactivation"

_genes, 2023, doi:10.3390/genes14061193_

Round 1
Reviewer 1 Report (New Reviewer)
The authors describe an interesting case of a female carrier with a heterozygous mutation in RS1 an skewed X-inactivation.
The paper is well written and includes a thorough analysis of the cases of manifesting females with mutation in RS1 including homozygous or compound heterozygous cases in consanguineous families and manifesting carriers in suspected skewed X-inactivation. From my part there are only few suggestions. The authors should indicate which OCT device was uses. The images are very nice with regard to the retinoschisis but the macula of the manifesting carrier should be shown at higher resolution to appreciate the comment of reduced lamination. Also, the macula is described as thickened but does this refer to the overall thickness or to the outer retina? Also, as the girl at the last examination wa 3 y and 7 months old, it would be nice to get some information on visual acuity, especially with respect to the OCT findings. Also, it would be nice to show the ERG data that are described to show reduced b-wave amplitudes (but not negative ERG) which is compatible with XLRS.
Author Response
The authors describe an interesting case of a female carrier with a heterozygous mutation in RS1 an skewed X-inactivation. The paper is well written and includes a thorough analysis of the cases of manifesting females with mutation in RS1 including homozygous or compound heterozygous cases in consanguineous families and manifesting carriers in suspected skewed X-inactivation.
We thank you for your time and kind comments.
From my part there are only few suggestions. The authors should indicate which OCT device was uses.
Included, please refer to line 69.
The images are very nice with regard to the retinoschisis but the macula of the manifesting carrier should be shown at higher resolution to appreciate the comment of reduced lamination.
OCT image has been adjusted accordingly.
Also, the macula is described as thickened but does this refer to the overall thickness or to the outer retina?
Refers to overall thickness; please refer to line 91 – 92.
Also, as the girl at the last examination was 3 y and 7 months old, it would be nice to get some information on visual acuity, especially with respect to the OCT findings.
Unfortunately, we do not have this information to include.
Also, it would be nice to show the ERG data that are described to show reduced b-wave amplitudes (but not negative ERG) which is compatible with XLRS
Included now as figure 3.
Reviewer 2 Report (New Reviewer)
Kirkby et al., in their article report a clinical case of a 2-year-old female with X- Linked Retinoschisis (XLRS) phenotype. XLRS is commonly seen in young male children, but this article adds to the existing knowledge of limited cases reported for female patients suffering from XLRS.
The article is very informative, well written, and describes in detail the unusual features of the XLRS phenotype seen in this patient. The finding of the “skewed” pattern of X-inactivation is valuable and is a great source of information for clinicians that deal with such rare cases of XLRS. As such, this article highlights the importance of testing for X-inactivation in such atypical forms of XLRS cases.
Author Response
Kirkby et al., in their article report a clinical case of a 2-year-old female with X- Linked Retinoschisis (XLRS) phenotype. XLRS is commonly seen in young male children, but this article adds to the existing knowledge of limited cases reported for female patients suffering from XLRS.
The article is very informative, well written, and describes in detail the unusual features of the XLRS phenotype seen in this patient. The finding of the “skewed” pattern of X-inactivation is valuable and is a great source of information for clinicians that deal with such rare cases of XLRS. As such, this article highlights the importance of testing for X-inactivation in such atypical forms of XLRS cases.
We thank the reviewer for their time and kind comments, which are greatly appreciated.
Reviewer 3 Report (New Reviewer)
Kirkby et al. presented a 2-year old female with XLRS. The authors provided clear and detailed ophthalmic examination data, molecular testing including X-chromosome inactivation assay. The authors then performed a comprehensive literature review on females presented with XLRS. The manuscript is well written and is scientifically sound. There are a few minor concerns to be addressed.
1. Line 102, p.(Tyr89fs36) is not properly annotated per HGVS guideline. The short form of p.Tyr89fs is fine, the detailed long form should include the altered residue. Please use variantvalidator.org if not certain on the annotation.
2. Line 103, please include the version no. of RefSeq transcript ID according to Supplemental Table 1.
3. It may be helpful to test father's sample for XCI and determine which X chromosome in proband is inherited from father. Likely it will show that the majority of father's allele is un-methylated, to further support the conclusion of this study.
4. It may be helpful to compile all the female cases with XLRS discussed in the Discussion into a Table, with molecular testing results and clinical presentation to support the following claim on line 249-251, "In these cases, where both alleles are affected, the retinal features are more consistent with typical XLRS, though wide variation in phenotype is evident, including within those affected by the same causative variant."
5. Line 190: "No variants of unknown clinical significance were found." perhaps should be "No variant of uncertain significance was not reported in the clinical report", given variants of uncertain significance are often found in panel tests, but not reported because they are not relevant to patient phenotypes.
6. Line 195: Please define "deep intronic variants". The reviewer would consider the RS1 variant c.53-34A>G "deep intronic" (HGMD), unless it is cleared defined to exclude this variant.
7. Line 241-246. No genetic testing was performed in Reference 23, thus "their presumed carrier mother and sister were unaffected.
Author Response
Kirkby et al. presented a 2-year old female with XLRS. The authors provided clear and detailed ophthalmic examination data, molecular testing including X-chromosome inactivation assay. The authors then performed a comprehensive literature review on females presented with XLRS. The manuscript is well written and is scientifically sound.
Thank you for your time and kind comments which are most appreciated.
There are a few minor concerns to be addressed. 1. Line 102, p.(Tyr89fs36) is not properly annotated per HGVS guideline. The short form of p.Tyr89fs is fine, the detailed long form should include the altered residue. Please use variantvalidator.org if not certain on the annotation.
Thank you. The text has been corrected with p.(Tyr89LeufsTer37) accordingly (see title, lines 132, 152, 203, and table 1).
- Line 103, please include the version no. of RefSeq transcript ID according to Supplemental Table 1.
The text at line 134 has been amended with "NM_000330.3"
- It may be helpful to test father's sample for XCI and determine which X chromosome in proband is inherited from father. Likely it will show that the majority of father's allele is unmethylated, to further support the conclusion of this study.
We agree that this would be useful. Unfortunately, the family live remote to the tertiary centre in which they were seen, and it would be very difficult to arrange this analysis in a timely fashion.
- It may be helpful to compile all the female cases with XLRS discussed in the Discussion into a Table, with molecular testing results and clinical presentation to support the following claim on line 249-251, "In these cases, where both alleles are affected, the retinal features are more consistent with typical XLRS, though wide variation in phenotype is evident, including within those affected by the same causative variant."
Thank you for this excellent suggestion. This has been included in table 1.
- Line 190: "No variants of unknown clinical significance were found." perhaps should be "No variant of uncertain significance was not reported in the clinical report", given variants of uncertain significance are often found in panel tests, but not reported because they are not relevant to patient phenotypes. 
Amended, as per line 205.
- Line 195: Please define "deep intronic variants". The reviewer would consider the RS1 variant c.53-34A>G "deep intronic" (HGMD), unless it is cleared defined to exclude this variant.
Amended, as per line 208 – 209.
- Line 241-246. No genetic testing was performed in Reference 23, thus "their presumed carrier mother and sister were unaffected.
Amended, as per line 261.
Reviewer 4 Report (New Reviewer)
The authors, Kirkby et al, describe a case of a 2-year old female with X-linked retinoschisis disease phenotype. This is an atypical case as the female is carrier for RS1 pathogenic variant, and the disease phenotype seems to be associated with disease variant manifestation due to skewed X-inactivation. This case report is useful for the field, and mostly this is a well-written manuscript. The comments on the manuscript are included below.
Major comments:
1. A brief introduction section is needed. In the current form, the manuscript directly starts with case report.
2. Methods section is missing. Several techniques like FAF, OCT, NGS, X-inactivation analysis etc. were used and these can be described in the Methods section. Even more important would be to include the information that blood sample was used for testing X-inactivation.
3. Skewed X-inactivation has been proposed as a major reason for the disease phenotype observed, and has been referred to in the title as well. However, in the clinical case report section, the result is not described well enough. Additionally, it would be important to show the data as a figure or a figure panel.
4. Lines 69-74 describe electrophysiologic information. It would be good to include a figure to show the traces for the waveforms measured.
5. Discussion can be improved a lot. A lot of text has been used to summarize the results of other publications (which can be shortened). Subsequently, the results have again been summarized too (lines 185-199) in the discussion. Thus, overall the discussion section has been used to summarize the authors’s work and other researchers’ work. However, it should be taken as an opportunity more to discuss how the published work relates )or does not relate) to what the authors’ found, and how it might be useful to the field, or what kind of further research could be performed.
6. The authors discuss consanguinity as a possible cause in some cases, however, the information whether the child’s parents had a consanguineous marriage or not has not been mentioned.
Minor comments:
1. Figure 1: The yellow asterisk used is difficult to see in the color fundus photograph. It might be better to use a higher contrast color for ease of view.
2. Figures D, E, and F are very useful but it is difficult to know which areas from C or D are focused on to show E and F. If possible, please highlight or use dotted lines to highlight the areas that were focused on for E and F. Additionally, the scale bars are also not visible, and thus it is not clear if all images are on the same scale or different scales. Please include legible scale bars on all images.
3. Line 53-54: “area with associated yellow material”. Do the authors mean lipofuscin?
4. Figure 3: The line showing OCT section is not clearly visible. It would be good to make it thicker.
5. In the figure legends, the word ‘asterix’ needs to be replaced with ‘asterisk’.
English used is good. The contents (such as the length and content of Discussion) can be improved.
Author Response
The authors, Kirkby et al, describe a case of a 2-year old female with X-linked retinoschisis disease phenotype. This is an atypical case as the female is carrier for RS1 pathogenic variant, and the disease phenotype seems to be associated with disease variant manifestation due to skewed X-inactivation. This case report is useful for the field, and mostly this is a well-written manuscript. The comments on the manuscript are included below.
We thank you for your time and comments, which are most appreciated.
Major comments: 1. A brief introduction section is needed. In the current form, the manuscript directly starts with case report.
Thank you for this suggestion. A brief introduction is now included lines 27 – 51.
- Methods section is missing. Several techniques like FAF, OCT, NGS, X-inactivation analysis etc. were used and these can be described in the Methods section. Even more important would be to include the information that blood sample was used for testing X-inactivation.
Expansion of methods have been included (including lines 69 – 69) and incorporated into the text. Information regarding blood sample for testing X-inactivation may be found in line 142.
- Skewed X-inactivation has been proposed as a major reason for the disease phenotype observed, and has been referred to in the title as well. However, in the clinical case report section, the result is not described well enough. Additionally, it would be important to show the data as a figure or a figure panel.
We have described the X-inactivation analysis in detail in lines 141 – 148. The results are expressed as ratio, and not available as a figure.
- Lines 69-74 describe electrophysiologic information. It would be good to include a figure to show the traces for the waveforms measured.
Included as figure 3.
- Discussion can be improved a lot. A lot of text has been used to summarize the results of other publications (which can be shortened). Subsequently, the results have again been summarized too (lines 185-199) in the discussion. Thus, overall the discussion section has been used to summarize the authors’s work and other researchers’ work. However, it should be taken as an  opportunity more to discuss how the published work relates )or does not relate) to what the authors’ found, and how it might be useful to the field, or what kind of further research could be performed.
We think it is important to describe the cases in the literature, and indeed refer to other reviewer’s positive comments in this regard who found these descriptions helpful (“a comprehensive literature review on females presented with XLRS,” and “well written and includes a thorough analysis of the cases of manifesting females with mutation in RS1”). This information is now additionally presented in a table as per another reviewer’s recommendation, to facilitate comparison to presented case. Please also refer to lines 220 – 227, and 267 – 271 for examples of relating our findings to published works.
- The authors discuss consanguinity as a possible cause in some cases, however, the information whether the child’s parents had a consanguineous marriage or not has not been mentioned.
Thank you, this was an important omission and the text has been amended in line 64.
Minor comments: 1. Figure 1: The yellow asterisk used is difficult to see in the color fundus photograph. It might be better to use a higher contrast color for ease of view.
Colour changes in figures 1 and 2.
- Figures D, E, and F are very useful but it is difficult to know which areas from C or D are focused on to show E and F. If possible, please highlight or use dotted lines to highlight the areas that were focused on for E and F. Additionally, the scale bars are also not visible, and thus it is not clear if all images are on the same scale or different scales. Please include legible scale bars on all images.
Figure 1 E and F now have an approximated mapping of location associated with each. Scale bars are located in all OCT imaging.
- Line 53-54: “area with associated yellow material”. Do the authors mean lipofuscin?
We have described the appearance as one is unable to be certain.
- Figure 3: The line showing OCT section is not clearly visible. It would be good to make it thicker.
Figure (now figure 4) has been adjusted accordingly with thicker lines.
- In the figure legends, the word ‘asterix’ needs to be replaced with ‘asterisk’.
Amended as per lines 75 and 86.
Round 2
Reviewer 4 Report (New Reviewer)
The authors have addressed all the comments/concerns. The addition of table is very useful for the readers.
This manuscript is a resubmission of an earlier submission. The following is a list of the peer review reports and author responses from that submission.
Round 1
Reviewer 1 Report
The authors present a very comprehensive clinical report of a 2 year old female affected with retinoschisis, an X-linked retinal disorder associated to RS1 (Xp22.2-p22.1) mutations and characterized by symmetric bilateral macular involvement with onset in the first decade of life. XLRS is inherited in an X-linked manner, so it generally affects males. Females who inherit a pathogenic variant will nearly always have normal visual function. Only female members of consanguineous families could show some XLRS clinical traits as they are homozygous for a pathogenic RS1 variant.
The proband reported in this work harbours a novel variant in the RS1 gene (c.266delA, p.Tyr89fs) in heterozygosity. The pathogenicity of this variant is not questionable. A single nucleotide deletion (c.266delA) causes a frameshift in the coding sequence of the gene and leads to the synthesis of a truncated protein. Besides, cosegregation studies show that the father (XLRS affected) harbours the same variant.
The clinical traits of the proband are very severe, particularly considering the thickening and morphology of the macula, the few cystic lesions in RPE atrophic areas and the age of onset of the disease. Overall the atypical retinal features (as defined by the authors) seriously question the involvement of a single disease-causing gene. It is much more likely that the clinical traits observed conform with a blended or multi mendelian phenotype, whose genetic factors have not been completely identified.
According to the authors, the genetic data to identify the gene causing the disease was obtained after studying the RP111 targeted gene panel. The genes (coding and non-coding regions) contained in the panel and the bioinformatics steps and criteria to identify the putatively pathogenic genetic variants have to be reported. The full data on the genetic variants identified, including VUS, should be added in the text.
To provide the reader with a more precise phenotype-genotype correlation the following points should be addressed:
- A second pathogenic RS1 allele could have been inherited from the mother. It should be reported if a full or partial RS1deletion, and splicing variants (NCSS and deep-intronic) were discarded.
- RS1 encodes a retinal cell adhesion protein that is assembled and secreted from photoreceptors and bipolar cells as a homo-oligomeric protein complex. Some of its interactors have been reported. They would be good targets for analysis and should be prioritized to identify putative pathogenic disease contributors. These data has to be included in the text.
- Following studies of families with overlapping phenotypes harbouring mutations in other genes besides RS1, such as CRB1 and RS1 (Khan et al.; ref 20 in the MS), and NR2E3 and RS1 (Al-Kuzaei et al.; ref 25 in the MS) the multi mendelian hypothesis should be deeply considered. It also applies to many syndromic and non syndromic retinal disorders. NGS methodologies are now available and affordable to pursue this aim.
To define phenotype-genotype correlations in hereditary retinal disorders is not a trivial task. But the methodological tools are now ready to successfully combine clinical and genetic data. A big leap in this field is more than needed to contribute to basic science, and to help the patient, clinicians and healthcare providers.
Author Response
We thank the reviewers for their comments, our responses are documented below:
The clinical traits of the proband are very severe, particularly considering the thickening and morphology of the macula, the few cystic lesions in RPE atrophic areas and the age of onset of the disease. Overall the atypical retinal features (as defined by the authors) seriously question the involvement of a single disease-causing gene. It is much more likely that the clinical traits observed conform with a blended or multi mendelian phenotype, whose genetic factors have not been completely identified.
Whilst we accept that we cannot rule out the involvement of other genes the case reported here shows similar phenotypic traits previously described in the literature in females with heterozygous variants.
According to the authors, the genetic data to identify the gene causing the disease was obtained after studying the RP111 targeted gene panel. The genes (coding and non-coding regions) contained in the panel and the bioinformatics steps and criteria to identify the putatively pathogenic genetic variants have to be reported. The full data on the genetic variants identified, including VUS, should be added in the text.
Text has been amended accordingly
To provide the reader with a more precise phenotype-genotype correlation the following points should be addressed:
- A second pathogenic RS1 allele could have been inherited from the mother. It should be reported if a full or partial RS1deletion, and splicing variants (NCSS and deep-intronic) were discarded.
There was no evidence for a deletion of the RS1 gene but we cannot rule out a deep intronic variant in RS1 but none have been described to date. The text has been amended to reflect only 1 pathogenic variant was found
- RS1 encodes a retinal cell adhesion protein that is assembled and secreted from photoreceptors and bipolar cells as a homo-oligomeric protein complex. Some of its interactors have been reported. They would be good targets for analysis and should be prioritized to identify putative pathogenic disease contributors. These data has to be included in the text.
The text has been amended to acknowledge this point but more work needs to be done to identify the specific interactors
- Following studies of families with overlapping phenotypes harbouring mutations in other genes besides RS1, such as CRB1 and RS1 (Khan et al.; ref 20 in the MS), and NR2E3 and RS1 (Al-Kuzaei et al.; ref 25 in the MS) the multi mendelian hypothesis should be deeply considered. It also applies to many syndromic and non syndromic retinal disorders. NGS methodologies are now available and affordable to pursue this aim.
The text has been amended to reiterate that no other potential disease-causing variants were identified
Reviewer 2 Report
This case report demonstrated a 2 years-old girl with X-linked retinoschisis. Although the clinical context was not typical for X-linked retinoschisis, it might be caused by random skewed X-inactivation.
Review
- Please provide transcript accession number.
- Please use variant rather than mutation throughout the manuscript.
- In line 87, please revise the term “infant”.
- In line 166, please revise “RS1 639delG mutation” to “RS1 c.639delG variant”
- In line 169, please revise “a novel 588-593ins.C” to appropriate HGVS nomenclature.
- Please provide pedigree figure in this family. Please provide fundus photo with OCT in her father. This will help the readers understand the case.
Author Response
We thank the reviewers for their comments, our responses are documented below:
1. Please provide transcript accession number
Added to text
2. Please use variant rather than mutation throughout the manuscript.
Text amended accordingly
3. In line 87, please revise the term “infant”.
Text has been amended
4. In line 166, please revise “RS1 639delG mutation” to “RS1 c.639delG variant”
Text has been amended
5. In line 169, please revise “a novel 588-593ins.C” to appropriate HGVS nomenclature.
Text has been amended
6. Please provide pedigree figure in this family. Please provide fundus photo with OCT in her father. This will help the readers understand the case.
Widefield retinal imaging of father included as figure 1. Unfortunately, unable to provide a pedigree nor OCT imaging.
Round 2
Reviewer 1 Report
The case presented by the authors shows a very severe phenotype of XL retinoschisis that affects a 2-year-old female. The point raised by the reviewer does not question the clinical report, but seriously questions the hypothesis that the clinical traits presented by the proband are only associated with the pathogenic variant identified in the RS1 gene. Phenotype-genotype correlations are very relevant to improve the knowledge of the molecular basis of the disease, help clinicians dealing with similar cases, establish a secure prognosis and guide the patient to suitable treatments. It cannot be ignored that after many years of genetic testing of retinal disorders, and huge methodological improvements, it is now known that severe retinal phenotypes not unfrequently appear to be associated with multimendelian inheritance. If NGS genetic testing was performed (gene panel containing 111 genes) all the genetic variants identified after gene targeted sequencing should be fully reported. And it is hard to believe that they were not detected. Without this data the considerations (p 6 discussion section) about epigenetic changes, comparison with homozygous or compound heterozygous females for RS1 variants and that XCI ratios can skew with age in some tissues (not known in the retina) remain unsolid arguments.
Round 3
Reviewer 1 Report
Although there is no question about the quality of the clinical report of the case presented, the genetic basis of the disease remain totally unmet. Besides the obvious association with RS1, the involvement of other genetic contributors, modifiers and interactors has been ignored. The authors have missed the oportunity to make a substantial contribution to the genetic basis of this disorder. It is also very questionable that NGS sequencing of the coding regions of 111 genes did not identify any VUS or pathogenic allele leading to reported or novel genetic candidates causing the disease. These data should have been included in the list.